# Rhizospheric soil chemical properties and microbial response to a gradient of *Chromolaena odorata*(L) invasion in the Mount Cameroon Region

Victor Nzengong Juru[1,2], Lawrence Monah Ndam [2,3,4]*, Blaise Nangsingnyuy Tatah[3], Beatrice Ambo Fonge[1,4]

**1** Department of Plant Science, University of Buea, Buea, Cameroon, **2** Institute for Nature, Health and Agricultural Research (INHAR), Buea, Cameroon, **3** Department of Agronomic and Applied Molecular Science, University of Buea, Buea, Cameroon, **4** Agroecology Research Group, University of Buea, Buea, Cameroon

* lawrencendam3@gmail.com

**Data Availability Statement:** All relevant data are within the manuscript and its Supporting Information files.

## Abstract

*Chromolaena odorata* is a noxious alien invasive weed species with an enormous impact on the terrestrial ecosystem. The allelopathic potentials of this weed have had little attention, leading to changes in soil properties and microbial communities. This study investigates the impacts of *Chromolaena odorata* invasion gradients on rhizospheric soil chemical properties and microbial response in the Mount Cameroon Region. Forty-eight soil samples at four different degrees of invasion (uninvaded, low degree invasion, moderate degree invasion and high degree invasion) based on species coverage within subplots in four study areas were collected and rhizospheric soil chemical properties, microbial load, phosphatases activities and secondary metabolites were evaluated. At medium-degree invasion, rhizospheric soil concentrations of P, K and Fe increased with arbuscular mycorrhizal fungi (AMF) colonization and phosphatases enzyme activities. Soil C, N and organic matter were significantly increased at high-degree invasion, supporting the use of the plant as a fallow crop. Acid phosphatase activity ranged from 0.69 to 0.90 mmol h$^{-1}$ kg$^{-1}$ and was significantly different at different degrees of invasion. AMF colonization ranged from 23.33 to 50.00%, with a strong positive correlation between AMF colonization and phosphatase activity. Soil bacterial load was high (46 × 10$^5$ CFU/g– 67 × 10$^5$ CFU/g), with mostly *Staphylococcus* having health concerns about its spread. The invasion situation had no significant effect on soil bacterial load, but high-degree invasion significantly increased fungal load. Low-degree invaded soils had high saponin (24.55±0.00 mg/g), flavonoid (47.7 mg/g) and tannin (28.68 mg/g) concentrations. The investigation reveals that *Chromolaena odorata* invasion altered rhizospheric soil properties and microbial communities significantly, thereby influencing ecosystem dynamics and soil nutrient availability. However, further studies elucidating kinds of secondary metabolites, identifying microbial communities, and monitoring soil changes influenced by C. odorata are essential for effective ecosystem management.

**Funding:** The author(s) received no specific funding for this work.

**Competing interests:** The authors have declared that no competing interests exist

## 1. Introduction

The globalisation of human activities has caused plants to spread beyond their natural habitats, resulting in a significant change in ecosystem function and structure and a change in soil chemical composition [1,2]. This biological invasion is the second-largest threat to biodiversity after habitat destruction and is one of the top most environmental concerns [3]. The increase in invasive species and their irreversible impact on ecosystems is a major concern for conservationists worldwide [4]. The slow and subtle changes caused by invasive species such as *Chromolaena odorata* may be difficult to detect until they have reached their consolidated state and exert maximum negative effects [5].

*Chromolaena odorata* (L) R. King and H. Robinson (Asteraceae) is a perennial shrub that originated from Central and South America [6]. However, it has become an invasive weed in tropical and subtropical regions of Africa and other parts of the world, spreading rapidly in open areas such as roadsides, riverbanks, disturbed forests, and plantations [7]. The weed exhibits a high reproductive capacity and has several common names in different regions in Cameroon, such as Siam weed, devil weed, French weed, communist weed, hagonoy, co hoy, triffid weed, Bitter bush, Christmas Bush, Camphur Grass, Common Floss Flower, and achacasala [8,9].

*Chromolaena odorata* is one of the world's worst invasive weeds that severely threaten agriculture, biodiversity and the environment. The plant releases allelochemicals such as flavonoids, terpenoids and alkaloids that alter nutrient fluxes and even modify soil microbial communities [10]. The weed outcompetes native plants via light competition and accumulation of soil-borne native pathogens and has a fast growth rate [11]. Although *C. odorata* has been used as a medicinal weed and fallow crop in the Mount Cameroon Region, studies have established that its eradication, containment, exclusion or population reduction is difficult due to the limited management interventions that tackle the general ecosystem [12]. Its proliferation in forest gaps, along riversides, roadsides and plantations results in the rapid recolonization of agricultural fields, causing significant economic losses [9].

Invasive species such as *Chromolaena odorata* release root exudates that stimulate soil microbial communities and increase nutrient cycling [13]. Invasive plants alter soil microbial community structure, which impacts native plant performance [14,15] by increasing pathogenic microbes and decreasing beneficial ones required for the fitness of native plants [16]. Invasive plants affect environments by altering the soil microbial community in their rhizosphere. For instance, low and high degrees of *Wedelia trilobata* invasion increase the richness of soil fungal communities but do not affect soil bacterial communities [17]. They can form arbuscular mycorrhizal fungal (AMF) associations or degrade local mycorrhizal fungi. This often favours the proliferation of generalist AMFs to local specialist ones [18]. Mycorrhizal fungi are crucial in optimising soil aggregation and are essential in water and nutrient uptake, including phosphorus, zinc and copper. In phosphorus-deficient tropical environments, these fungi participate in phosphorus fixation, comprising phosphorus recovery in the soil profile [19].

The rate of phosphorus mineralisation is governed by microbial activity and plant-synthesised enzymes [20]. However, some invasive plant species can boost the activity of phosphatase enzymes in rhizosphere soil due to the higher phosphorus uptake [21]. Phosphatases are a class of enzymes that break ester-phosphate bonds, resulting in the release of phosphate (P) that can be taken up by plants or microorganisms [22]. Moreover, the activities of phosphatases, including acid and alkaline phosphomonoesterases, are influenced by soil properties, soil organism interactions, plant cover, and leachate inputs [20]. *Pseudomonas sp.* produces phosphatase that stimulates enzyme synthesis by intact roots, releases inorganic phosphate from

organic and complex inorganic materials and plays an essential role in the phosphorus cycle [23]. It also solubilises organic and inorganic phosphates, thereby supporting plant growth [24].

Proper management of the invasive species such as *Chromolaena* will depend on a precise evaluation of all stages of the invasion process in natural ecosystems [25]. Furthermore, more studies are necessary to evaluate the plant's impact on below-ground biodiversity and its ecological ramifications on restoration activities in invaded ecosystems. Although previous studies have focused on the plant's general effect on soil chemical and physical properties, more attention needs to be given to the soil and microbial interaction within the rhizosphere to establish the plant's complete impact [26]. Finally, there is a need for more data on the allelopathic potential of *C. odorata* at different degrees of invasion and their effects on soil microbial communities, phosphatase activities during invasion, as noted by [10]. Understanding the effects of *C. odorata* invasion on soil properties and microbial communities is critical in the development of better and more effective management strategies aimed at mitigating the impacts of invasive species and maintaining the ecological health of the Mount Cameroon region. Hence, this study aimed to assess the impact of *C. odorata* invasion on soil chemical properties, microbial load and rhizospheric biological activities in the Mount Cameroon Region. The study specifically assessed the effects of different degrees of *C. odorata* invasion on soil chemical properties, soil microbial load, and phosphatase activities. It also analysed the secondary metabolites released to rhizospheric soils at different degrees of invasion.

## 2. Materials and methods

### 2.1 Site description

The Mount Cameroon Region (MCR) is found in the Southwest region of Cameroon and covers a surface area of 5695.5 km$^2$, stretching from latitude 3°57' to 4°28' N and longitude 8$^0$58' to 92°4' E [27]. The region stretches about 20 m above sea level to a height of 4100 m, called Mount Cameroon, which is situated at 4°13' N, and 9$^0$10' E. This is the highest point in West and Central Africa. The climate is humid with annual rainfall ranging from 2085 mm to 10000 mm. The mean temperature is 25°C, which decreases by 0.6°C per 100 m ascent [28]. According to the Institute of National Statistics of Cameroon (2010), the population of the Southwest region is 1384300, with about 400,000 people living around the Mount Cameroon National Park area [29]. The region hosts several plantations owned by the Cameroon Development Cooperation (CDC) and small scheme holders.

This study was conducted in Buea, Bakingili, Limbe and Idenau localities around the Mount Cameroon National Park. Field surveys were done in the two major seasons between September 2019 and November 2020 from three land uses; roadsides, farmlands and forests. Accessibility, tourism, anthropogenic activities, historical facts and safety were paramount in site selection. The study investigated an invasive plant species, regarded as a weed, in publicly accessible areas of the Mount Cameroon region through non-invasive observations and experiments. As it did not involve environmental manipulation or specimen collection for commercial purposes and aimed to enhance ecological understanding and management, no formal permits were necessary under local regulations.

### 2.2. Experimental design

The modified Whittaker method [30] was used to lay 120 permanent plots of size 20x50 m, with 10 plots per land use (roadside, farmland and forest) and 30 per site. Along the inside border, subplots of size 2x5 m were placed systematically in alternate corners and a 5 x 20 m subplot was placed at the centre. All plots with *C. odorata* were tagged and noted. In April 2022,

rhizospheric soil samples were collected at different degrees of invasion from the four sample areas using slightly modified methods [31]. Each sample area was divided into four sites based on the degree of *C. odorata* invasion, using coverage of the species within each subplot as (i) uninvaded (0%, ND), (ii) low degree (<35%, LD), (iii) medium degree (50%, MD) and (iv) high degree (>75%, HD). Forty-eight soil samples were collected at a depth of 5 cm using a soil auger around the root zone of *C. odorata* from the four study areas at the four different degrees of invasion. The soils were bulked based on the different treatments, air dried, kept in sealed polythene bags and stored at about 10°C. The soil samples were collected from Buea (N 04°7′52.97"/E 09°13′24.27"to N04°12′27.42"/E 09°16′20.31") and altitude 781–971 m above. sea level), Limbe (N 03°35′59.59"/E09°15′14.28"to N04°2′10.25"/E09°5′53.25") and altitude 63–181 m above sea level), Bakingili (N 04°4′11.89"/E 09°2′11.93"to N 04°3′3.56.99"/E 09°1′50.77") and altitude 9–24 m above sea level and Idenau (N04°14′21.82"/E08°59′45.91"to N 04°12′13.32"/E08°59′53.95") and altitude 9–52 m above sea level. All soil samples were collected by purposive random sampling, targeting the soils around the root zone and were homogenised based on the degree of invasion and study area (4 study areas x 4 degrees of invasion), resulting in sixteen treatments, which were taken to the laboratory for analysis after sieving with a 2 mm sieve.

## 2.3. Soil chemical properties

Standard methods were used to analyse soil chemical properties, including pH, moisture %, organic matter, organic C, N, Mn, P, Fe, K, Ca and Mg. The pH was measured using the Thermo-Russell pH meter following the method of Black [32], where 10 g of 2 mm soil samples were weighed into beakers, 25 mL of KCl and distilled water were added, and a soil solution was obtained. The solution was stirred and left overnight, and the pH was measured the following day by dipping the calibrated electrode into the sample. Total nitrogen was determined by colourimetry from absorbance $\lambda = 650nm$ using a spectrophotometer. Soil samples were digested at 370°C with concentrated $H_2SO_4$ in the presence of sodium sulphate/selenium catalyst (Kjeldal method) and absorbance was used to determine total nitrogen against a standard solution [33]. Organic carbon was determined by the Walkley-Black method, where the organic carbon in the soil was oxidised with $K_2Cr_2O_3$ and concentrated $H_2SO_4$ and the colourimetric readings were compared with the blank (sucrose and water) [32]. Available phosphorus in the soil was determined colourimetrically using the ammonium molybdate blue colouration method. Mg and other bases (Ca, K and Na) were extracted using ammonium acetate and determined colourimetrically on a spectrophotometer using the "Titan yellow method". The concentrations of Mn, Fe, $Na^+$, $K^+$ and $Ca^{2+}$ were determined using the Gallenkamp flame photometer by way of Atomic Emission Spectrometers, involving the thermal excitation of free atoms.

## 2.4 Microbial load

**2.4.1 Bacterial count.** To obtain the bacterial count of soil samples, one gram of soil sample was mixed with 10mL of distilled water and serially diluted nine times ($10^{-1}$ to $10^{-9}$) by adding sterile distilled water to each test tube, homogenising the preparations at each step. From the preparations, 1 mL of soil sample was mixed with 9 mL of sterile distilled water to make 9-fold serial dilutions of each preparation, and the serial dilutions were made up to $10^9$ dilutions. A volume of 0.1 mL from the seventh ($10^7$), eighth ($10^8$), and ninth ($10^9$) dilutions were each inoculated in duplicate on a nutrient agar plate using the spread plate technique. The inoculated plates were incubated at a temperature of 37°C for 24 hours. The same serial dilutions were made for other soil samples and they were also incubated at the same

temperature and time. After incubation, the bacterial colonies formed on the nutrient agar media were counted and recorded, and the plates were labelled with the dilution number. The total number of bacteria per mL of the original soil sample was calculated by multiplying the number of colonies counted by the dilution factor (Eq 1) [34]. To calculate the colony-forming units (CFU) of bacteria per gram of soil, the number of colonies was multiplied by the reciprocal of the dilution factor (Eq 2).

$$\text{Total number of bacteria per mL} = \text{number of colonies counted} \times \text{dilution factor} \qquad (1)$$

$$\text{CFU/g} = \text{colonies number} \times 10^4 \qquad (2)$$

*2.4.1.1. Bacteria culture, phenotypic and biochemical characteristics.* The working space was cleaned and disinfected with 75% alcohol. Using a balance, 1 g of soil sample was weighed and added to a test tube containing 10 mL of sterile 0.9% NaCl. The test tube was shaken thoroughly to separate the bacteria from the soil particles. Using a micropipette and a sterile tip, 1000μL of the soil solution was transferred to the tube labelled $10^0$ dilutions. It was vortexed for 30 seconds to mix the bacterial cells into the saline thoroughly. This was followed by $10^{-1}$, $10^{-2}$ and $10^{-3}$ serial dilutions. This procedure was done for all 4 samples. Nutrient agar was prepared by measuring 14.0 grams into 500ml of water and autoclaved at 121˚C and at 15 lbs pressure per sq. in. for 15 mins. The soil sample dilutions were poured into sterile petri dishes and labelled. The agar was removed from the autoclave and allowed to cool in a water bath set at 50˚C, then poured into the petri dishes containing the samples and allowed to solidify by the pour plate technique. The petri dishes were incubated at 37˚C for 24 hours, after which the petri dishes were removed from the incubator, and the phenotypic characteristics were recorded. Catalase test was carried out by scraping a bacterial colony from the agar on a slide, 1 drop of hydrogen peroxide was added and results were recorded.

Gram staining was carried out using a clean, grease-free slide. A smear of suspension was placed on the clean slide with a loopful of samples. The slide was air-dried and heat-fixed. Crystal violet was added and kept for a minute and rinsed with distilled water. It was flooded with the gram's iodine for 1 minute, washed with 95% alcohol for 20 seconds and rinsed with distilled water. Safranin dye was added for 1 minute and washed with distilled water. The preparation was air-dried blotted dry and observed under the microscope.

**2.4.2. Fungal count and culturing.** The workspace was cleaned and disinfected with 75% alcohol. Potato dextrose agar (PDA) was prepared by weighing 5 g of agar into 100 mL of distilled water in a 150ml conical flask. The prepared agar was autoclaved at 121˚C and at 15 lbs pressure per sq.in. for 15 mins and serial dilutions were made for the 4 soil samples. The dilutions $10^{-3}$ to $10^{-5}$ were used. To calculate the colony-forming units (CFU) of fungus per gram of soil sample, the number of colonies was multiplied by the reciprocal of the dilution. The CFU/g was calculated as in Eq 2

The soil sample was measured in a 10 mL graduated cylinder and 1mL of the sample was introduced into sterile petri dishes and labelled. The agar was removed from the autoclave and allowed to cool. It was poured into the petri dishes containing the sample and left to solidify. The culture was kept in a cool, moist area for two weeks for fungal growth [35]. After two weeks the phenotypic characteristics were recorded and microscopy was done by putting a fungal sample of on the slide and stained with methylene blue then viewed under the microscope.

*2.4.2.1. Determination of arbuscular mycorrhizal colonisation.* Triplicate samples of fine roots of *C. odorata* were collected from each degree of invasion and each site, placed in sealed bags and promptly transported to the laboratory. Assessment of arbuscular mycorrhizal fungi

(AMF) root colonization was conducted according to [36]. Segments of root samples measuring 1–2 cm were placed in a 5% KOH solution for 24 hours at room temperature, rinsed three times with water on a fine sieve, acidified in 10% HCl (v/v) for 15 minutes and stained with 0.01% (w/v) fuchsine acid for 24 hours at room temperature. Random root segments of stained samples were selected and three replicates of 10 roots per slide were assessed for the presence of AMF structures (vesicles, arbuscules and hyphae) using an optical microscope (Biological compound microscope with replaceable LED light, OMAX 40X-2500X Trinocular, Germany). The mycorrhizal frequency (F %) was obtained as the ratio of colonised root fragments to the total number of observed root fragments.

## 2.5. Determination of phosphatase activities

The method for quantifying rhizosphere acid phosphatase activity followed [36]. Five plants were randomly selected at each invasion degree for acid phosphatase activity assessment. A spade was used to dig approximately 5 cm around each plant at a depth of 20 cm. One gram of root-adhering soil was collected from each plant and pooled to create a composite sample of 5 g. A microcentrifuge tube was filled with 1 g of the composite sample, to which 0.5 mL of 100 mM phosphate buffer was added, followed by 10 mM of P-nitrophenyl phosphate (p-NPP) in 100 μL solution. The final reaction mixture volume was adjusted to 1 mL with distilled water and the tube was vortexed for 2 minutes at room temperature and incubated at 37˚C for 1 hr with shaking (100 rpm). After incubation, the samples were centrifuged at 10,000 rpm for 5 min, and the clear supernatants were transferred to clean test tubes and treated with 2 mL 1 M NaOH. The resulting yellow-coloured filtrate was analysed using a colourimeter (Klett colourimeter, Clinical model, 800–3, 115 VAC) at λ = 430 nm. The concentration of soluble protein in the supernatant was measured using the Bradford reactive procedure [37] and the specific activity of the soil acid phosphatase was estimated in mg h$^{-1}$ g$^{-1}$ using the formula:

$$Protein\ content = \left(\frac{10 \times C}{17800}\right) \quad\quad (3)$$

Where 10 = constant, C = concentration of soluble protein and 17800 is the molecular extension coefficient of phosphatase

## 2.6. Secondary metabolites in rhizospheric soil (Quantitative Allelochemicals)

**2.6.1. Determination of total phenolic content (TPC).** The Folin-Ciocalteu colourimetric method [38] was employed to determine the total phenolic content of the aqueous soil samples. Specifically, 0.2 mL of soil sample was mixed in a test tube with 1.5 mL of Folin-Ciocalteu reagent and incubated for 5 minutes at room temperature. 1.5 mL of a 6% sodium carbonate solution was then added to the mixture, which was re-incubated for 90 minutes at room temperature. A quartz cuvette at 725 nm measured the absorbance of the resulting blue colour. Meanwhile, gallic acid standards were prepared by dissolving 0.75 mg of gallic acid in 5 mL of distilled water. Different volumes (0, 0.1, 0.2 and 0.3) of the gallic acid standard were pipetted into four labelled test tubes and distilled water was added to each to make up the volume to 0.3 mL. A 2.25 mL of Folin-Ciocalteu reagent was added to each test tube, followed by incubation for 5 minutes at room temperature. The mixture was re-incubated for 90 minutes at room temperature after adding 2.25 mL of 6% sodium carbonate solution. A quartz cuvette was used to read the absorbance of the resulting blue colour at 725 nm. This was used to plot a standard curve from which the total phenolic content of the samples was estimated. The total phenolic content of the samples was expressed as μgGAE/g soil sample (Fig 1).

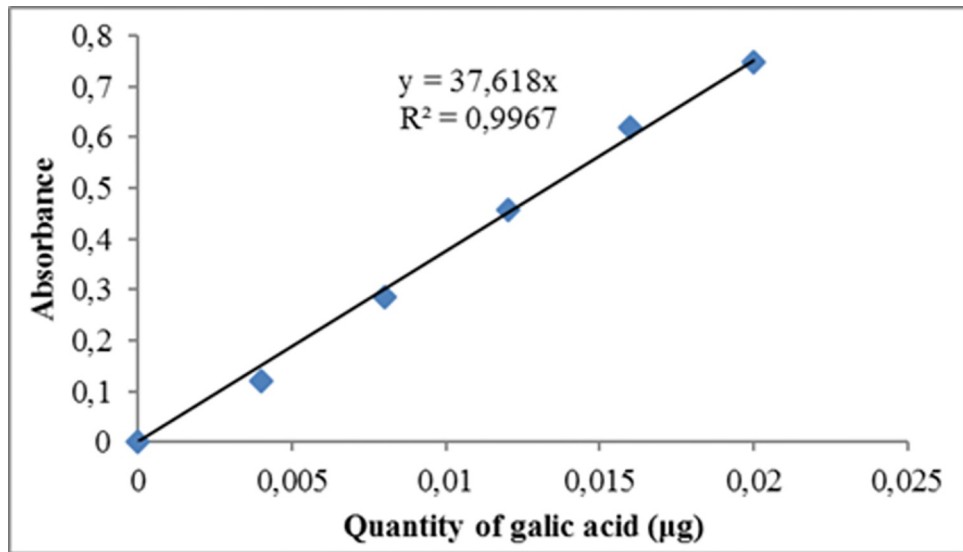

**Fig 1. Calibration curve of gallic acid in the determination of Phenolic content of samples.**

**2.6.2. Total flavonoid content.** Specifically, 0.15 g of soil sample was homogenised with 10 mL of an extraction solvent containing methanol, distilled water, and acetic acid (in a 14:5:1 v/v ratio). The mixture was filtered using Whatman No. 1 filter paper. 0.2 mL of the sample solution was then mixed with 1.8 mL of distilled water and 1 mL of aluminium chloride reagent consisting of 133 mg of aluminium chloride crystals and 400 mg of sodium acetate dissolved in 100 mL of the extraction solvent. The resulting mixture was vortexed and the absorbance was read at 430 nm against a blank. The amount of flavonoids was calculated using a standard solution of rutin (0.1 mg/mL) and the results were expressed in milligrams of Quercetin per gram of soil sample (Fig 2).

**2.6.3. Tannin content.** To determine the tannin content in the soil samples, the spectrophotometric method involving acidified vanillin was employed. One gram of each soil sample was mixed with 15 mL of acetone (10% acetic acid) and stirred for 15 minutes before filtering

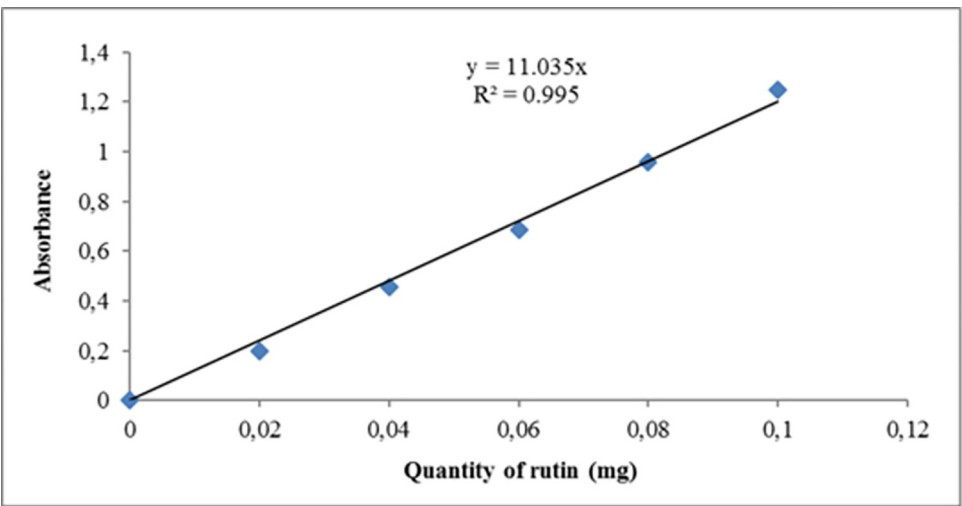

**Fig 2. Calibration curve of rutin in the determination of flavonoid content of samples.**

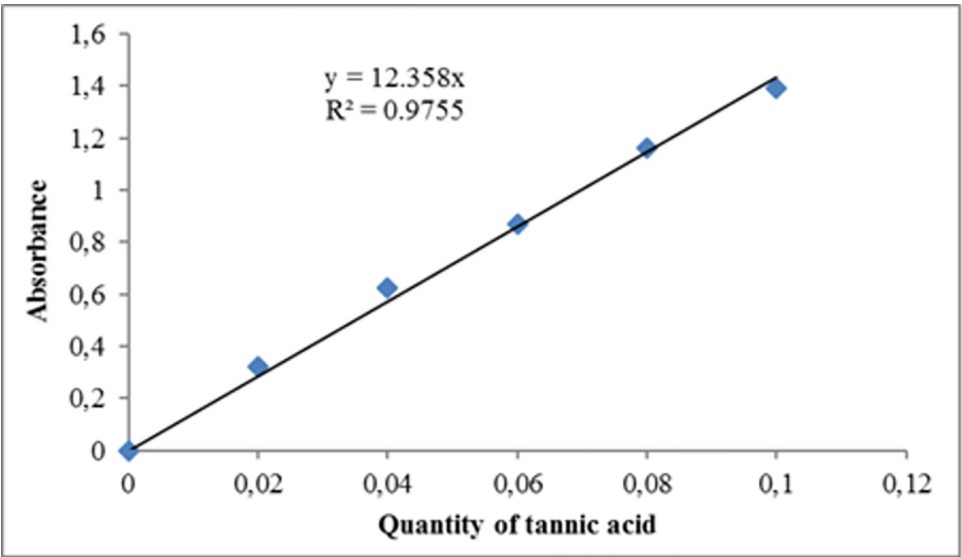

**Fig 3. Calibration curve of tannic acid for determination of total tannins.**

and diluting 20 times with distilled water. 1 mL of the diluted mixture was then added to each test tube wrapped with aluminium foil to exclude light. Then, 3 mL of a freshly prepared solution of 4% vanillin in ethanol (w/v) was added, and the mixture was stirred. Following this, 1 mL of concentrated HCl was added to each tube and the solution was kept at room temperature for 15 minutes before reading the absorbance at 500 nm against a blank. The amount of tannins was determined using a standard solution of tannic acid (5 mg/mL) and the results were expressed in milligrams of tannic acid per gram of soil sample (Fig 3).

**2.6.4. Saponin content.** The method used in this assay was based on the oxidation of triterpene saponins using vanillin, with strong sulphuric acid as the oxidising agent to break apart the aglycone from the complex saponin molecule. The estimated total saponin content in the soil samples, was by the method described by [39], which involved reacting the saponins with a sulphuric acid-vanillin reagent. Specifically, 0.5 mL of the sample was added to a Falcon tube containing 0.5 mL of vanillin (8% w/v) and 4.0 mL of sulphuric acid (72% w/v). The mixture was incubated at 60˚C for 15 minutes in a water bath and later cooled in an ice bath for 10 minutes. The absorbance of the sample was read at 560 nm, using aescin as a reference standard. The total saponin content in mg/g of soil sample was estimated using the calibration curve for aescin (Fig 4).

## 2.7. Data analysis

All data sets were analysed using the statistical software package IBM SPSS Statistics 25 for Windows. The dependent variables were subjected to univariate analysis of variance (ANOVA, $P<0.05$) to test the effect of treatments as categorical predictors. Significantly means were separated by the Posthoc Tukey Multiple Range test ($P<0.05$).

## 3. Results

### 3.1. Rhizospheric soil chemical properties

Soil P concentration ranged from 66.38–107.47 mg/kg with the highest concentration recorded under low degree invasion and the highest recorded under medium degree invasion.

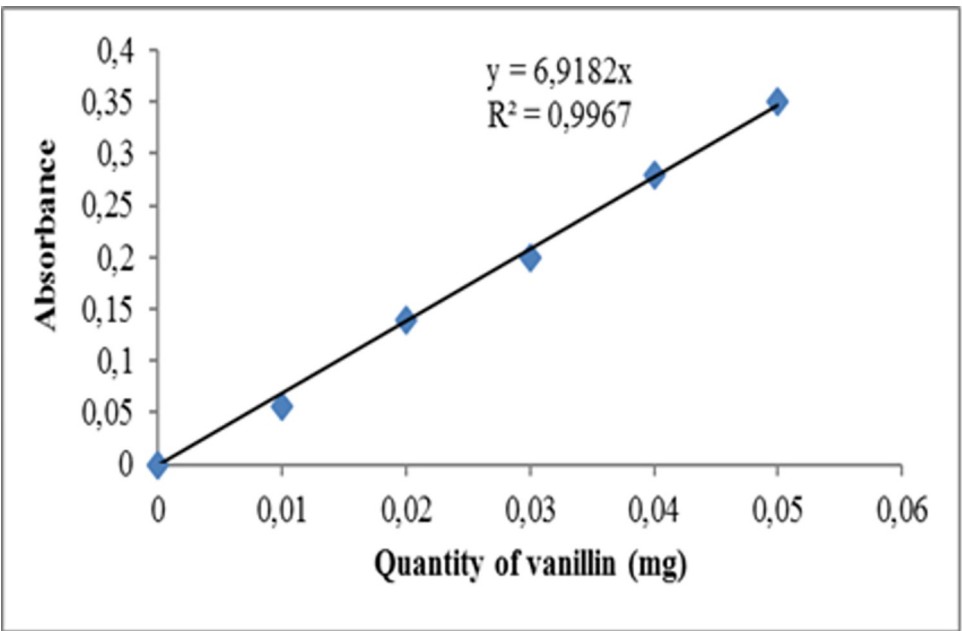

**Fig 4. Calibration curve of vanillin in the determination of saponin content of samples.**

Sites that had a low degree invasion therefore accounted for a 38. 1% increased P concentration, as compared to the uninvaded sites which were significantly different ($P<0.05$) across the various degrees of invasion. No significant increase was recorded in the P concentration at the uninvaded, low degree and high degree invasion sites (Table 1). The highest (3627.79 mg/kg) soil K concentration was recorded under low degree invasion by *C. odorata* and the lowest (2722.03 mg/kg) was recorded under high invasion, accounting for a 25% increase in K concentration, which differed significantly ($P<0.05$) across the various degrees of invasion. The K content for the uninvaded and that for the high invasion was statistically similar. The same observation was made for the low degree and medium degree invasion sites (Table 1). The calcium content ranged from 222.57 (low degree invasion)– 286.50 mg/kg (uninvaded) which significantly differed ($P<0.05$) across the various invasion degrees. However, the calcium concentration at low degree invasion was statistically similar to medium degree and that for high degree invasion was similar to the uninvaded sites (Table 1). The low degree invasion sites by *C. odorata* significantly increased ($P<0.05$) the Mg concentration, with a 57.25% increase as compared to the uninvaded sites which recorded the lowest concentration of 53.58 mg/kg. The highest Fe concentration (3728.20 mg/kg) was recorded on the uninvaded sites by *C. odorata*

**Table 1. Chemical properties (mineral salts) of soil under different degrees of *C. odorata* invasion.**

| Invasion Degree | Phosphorus (mg/kg) | Potassium (mg/kg) | Calcium (mg/kg) | Magnesium (mg/kg) | Iron (mg/kg) | Manganese (mg/kg) |
|---|---|---|---|---|---|---|
| Uninvaded | 66.57 ± 7.64**b** | 2834.20 ± 135.49**b** | 286.50 ± 69.96**a** | 53.58 ± 22.45**c** | 3728.20 ± 648.59**a** | 0.23 ± 0.03**a** |
| Low degree | 66.38 ± 10.22**b** | 3627.79 ± 153.33**a** | 222.00 ± 23.58**b** | 125.34 ± 12.95**a** | 2626.15 ± 524.05**b** | 0.24 ± 0.01**a** |
| Medium degree | 107.47 ± 14.88**a** | 3412.12 ± 362.23**a** | 224.57 ± 17.84**b** | 94.01 ± 33.88**b** | 3020.20 ± 137.37**b** | 0.25 ± 0.02a |
| High degree | 71.76 ± 12.14**b** | 2722.03 ± 137.88**b** | 254.95 ± 12.11**ab** | 100.07 ± 12.03**ab** | 1671.41 ± 112.77**c** | 0.24 ± 0.01**a** |

The values in the table are means and standard deviations of the four areas with the same degree of *C. odorata* invasion. Values with different letters in a column indicate significant differences ($P<0.05$). Units are given in mg/kg of sample.

while the lowest (1671.41 mg/kg) was recorded under high degree invasion by *C. odorata*. Therefore, the uninvaded sites increased the Fe content by 55.2% compared to the high degree invasion (Table 1). The different levels of invasion by *C. odorata* did not significantly affect the rhizospheric Mn concentration.

The high degree invasion by *C. odorata* increased the organic carbon concentration by 29.8%, as compared to the uninvaded, which ranged from 3.36 (uninvaded)– 5.45% (high degree) and differed significantly ($P$<0.05) from each other. The soil organic carbon concentration for the low degree invasion sites was statistically similar to the medium degree and the high degree invasion sites (Fig 5). The nitrogen concentration ranged from 0.17 (low degree)– 0.28% (high degree). The sites with high degree of invasion by *C. odorata* significantly increased the nitrogen concentration by 39.3% compared to the less invaded sites. The nitrogen concentration for the uninvaded sites was statistically similar to the sites that had a low degree of invasion by *C. odorata* and that for the medium degree of invasion was statistically similar to the sites invaded at high degree by *C. odorata*. (Fig 5). The highest organic matter concentration (9.19%) was recorded in sites that were highly invaded by *C. odorata* and the lowest concentration (5.82%) was recorded in the uninvaded sites. Therefore, high degree invasion by *C. odorata* increased the organic matter concentration by 36.7%. The different levels of invasion by *C. odorata* did not significantly affect the rhizospheric soil pH. The same observation was made for the moisture content (Fig 5).

## 3.2. Bacterial and fungal load

Bacterial colonies were present in rhizospheric soils of all the different invasion situations of *C. odorata*, with counts ranging from $46 \times 10^4$ CFU/g at uninvaded soil samples to $67 \times 10^4$ CFU/g in high degree invaded soils. The invasion of *C. odorata* however did not have a significant difference on the bacterial count of the rhizospheric soil samples at different invasion degrees (Fig 6A). Fungal count ranged from $10^3$ CFU/g in uninvaded soils to $3 \times 10^4$ CFU/g in high-degree invaded soils. The total fungal count varied with the degree of invasion and a high degree of invasion significantly increased the fungal load in the rhizospheric soil ($P$<0.05) with the high degree of invasion sites by *C. odorata* accounting for a 96.4% increase, as compared to the uninvaded sites (Fig 6B).

Bacterial shapes were mainly round, filamentous and irregular. The form was circular, some with rhizoids and some irregular with entire, undulate, or lobate margins. The colour of the colonies was cream in uninvaded soil samples and white in all degrees of invasion. All samples tested positive for the catalase and gram tests (Table 2).

These various bacterial shapes were not strongly distributed based on invasion degree, meanwhile, the main types observed under the microscope were *Staphylococcus*, *Corynebacterium*, *Bacillus* and *Cocci bacillus*. Based on phenotypic characteristics, samples from all degrees

**Table 2. Different bacterial colonies, their phenotypic and biochemical characterisations, shapes under the microscope and types present in rhizospheric soils at different degrees of *C. odorata* invasion.**

| Invasion situation | No of colonies | Shapes under the microscope | Catalase test | Gram test | Types of Bacteria |
|---|---|---|---|---|---|
| Uninvaded | 2 | Cocci in clusters and chains<br>Streptobacillus in chains | Positive | Purple | *Staphylococcus*<br>*Corynebacterium* |
| Low degree | 3 | Cocci in chains and clusters<br>Streptobacilli in pairs and chains<br>Cocci bacilli in pairs and clusters | Positive | Purple | *Staphylococcus*<br>*Corynebacterium*<br>*Cocci bacillus* |
| Medium degree | 1 | Spore forming rods | Positive | Purple | *Bacillus* |
| High degree | 2 | Spore-forming rods in chains<br>Cocci bacilli in clusters, pairs and single | Positive | Purple | *Bacillus*<br>*Cocci bacillus* |

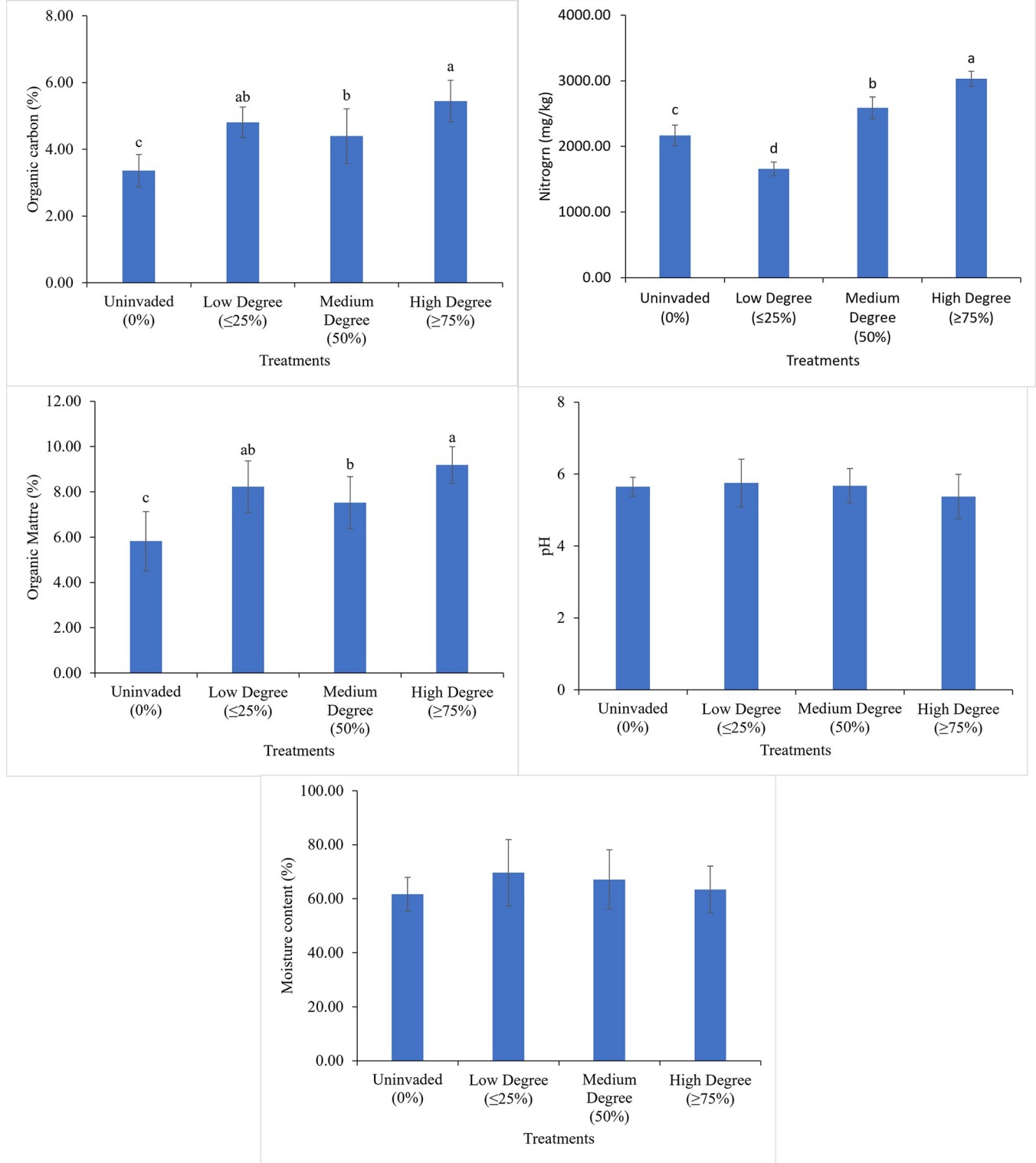

**Fig 5. Effects of different degrees of *C. odorata* invasion on rhizospheric soil organic carbon concentration, nitrogen concentration, organic matter concentrations, pH and moisture content.** Mean values with different indicate significant differences ($P<0.05$).

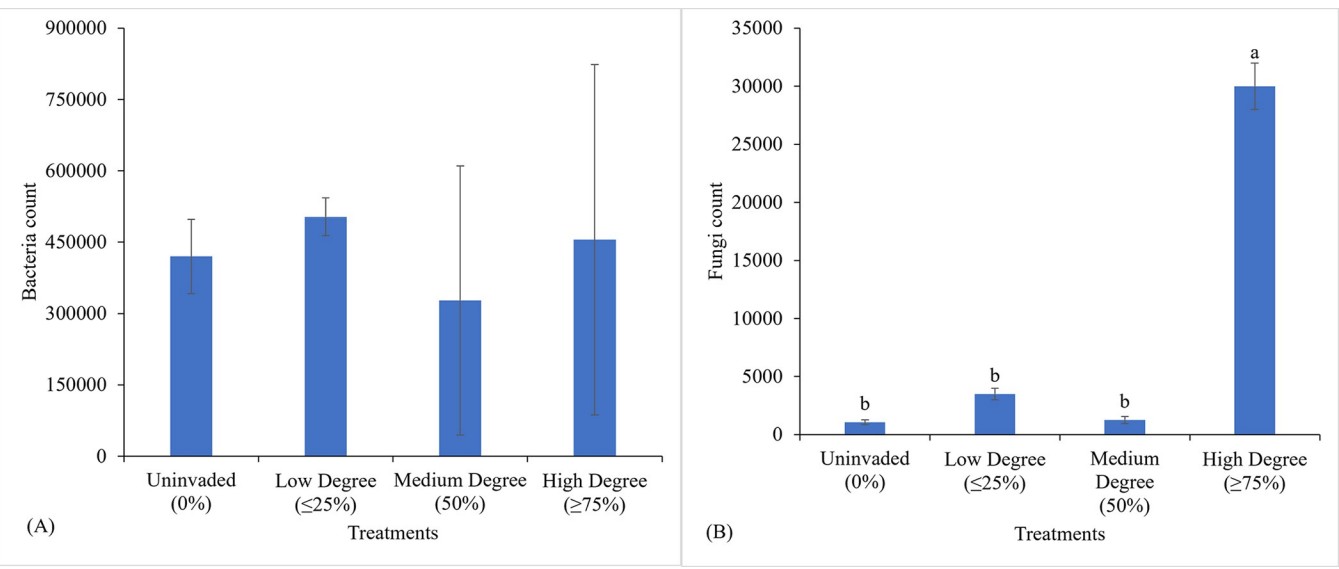

**Fig 6.** Effects of different situations of *C. odorata* invasion on bacterial load (a) and total fungal count (b) in rhizospheric soil samples. Mean values with different indicate significant differences (*P*<0.05).

of invasion had growth of fungi that spread, with very raised elevation, white hyphae, black spores, filamentous forms and translucent opacity. Under the microscope, all cultures had sexual and asexual spores, non-septate hyphae, cell walls, rhizoids, and mycelia and were mould-type. All degrees of invasion had the growth of *Rhizopus nigricans* fungi.

### 3.3. Mycorrhizal colonization

The root colonization of *C. odorata* by mycorrhizal ranged from 23.33 to 50.00%, with the highest Arbuscular mycorrhizal fungal (AMF) colonization recorded in root samples from low-degree invasion and the lowest in samples from high-degree invasion. Root colonization by AMF varied significantly (*P*<0.05), with the degree of invasion (F = 5.79), with low-degree invasion resulting in a significant increase in root colonization (Fig 7).

### 3.4. Acid phosphatase activities

The acid phosphatase activity in the rhizosphere of *C. odorata* ranged from 0.8952 (medium degree invaded sites) to 0.6857 mmol h$^{-1}$ kg$^{-1}$ (high degree invaded sites) which differed significantly from each other (*P*<0.05). The medium degree invaded sites increased the acid phosphatase activity by 30.5% compared to the high degree invaded sites. Also, the acid phosphatase activity of the uninvaded sites was statistically comparable to the medium degree invasion and both invasion degrees differed significantly from the low degree invasion sites (Fig 8).

### 3.5. Correlation between AMF root colonization and acid phosphatase activity of *C. odorata* at different degrees of invasion

There was a positive correlation (*r* = 0.69, *P*<0.05) between *Arbuscula*r mycorrhiza root colonization and acid phosphatase activity in the rhizospheric soil of *C. odorata*. The rhizospheric soils with increased mycorrhiza root colonization at varying degrees of invasion had higher acid phosphatase activity (Fig 9).

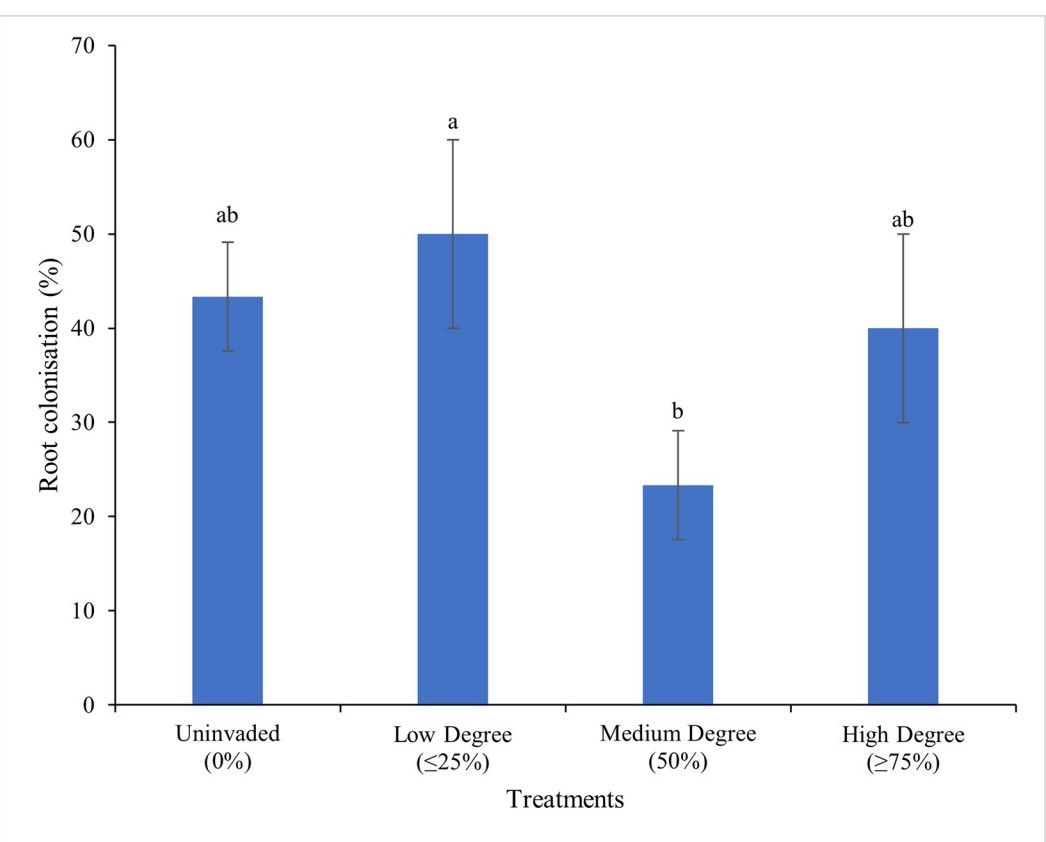

**Fig 7. Effect of different degrees of invasion on the arbuscular mycorrhiza root colonization (%) of *C. odorata*.** Data (Mean ± SD) with different letters are significantly different (Tukey's HSD, $P<0.05$).

### 3.6. Secondary metabolites

Flavonoids concentration ranged from 0 (high degree invasion)– 55.63 Rutin/g (uninvaded sites). High degree invaded sites reduced the flavonoids concentration by 100% as compared to the uninvaded sites which was significantly different ($P<0.05$) across the various degrees of invasion. The highest (31.98 mg/g) saponins concentration was recorded in the uninvaded sites and the lowest (7.88 mg/g) was observed under high degree invasion. The same trend was recorded in the tannin concentration where the high invaded sites by *C. odorata* reduced the tannins concentration by 75.4%, which differed significantly ($P<0.05$) across the various degrees of invasion. The tannins concentration ranged from 20.16 (high degree invasion)– 32.66 mg/TAE/g (uninvaded). The high invaded sites by *C. odorata* reduced tannins concentration by 38.3% which differed significantly ($P<0.05$) across all the degrees of invasion. The medium degree of invasion sites recorded a significantly higher concentration of total phenols compared to the other sites, where the lowest was recorded in the sites that were less invaded by *C. odorata*. Therefore, medium degree invasion accounted for a 57.8% increase in total phenol concentration as compared to the low degree invaded sites (Fig 10).

Significantly, very strong positive correlations were recorded between saponins, flavonoids and tannins in the rhizospheric soil of *C. odorata* (Table 3). Negative correlations which were not significant were recorded when the total phenolic compounds were correlated with saponins, flavonoids and tannins.

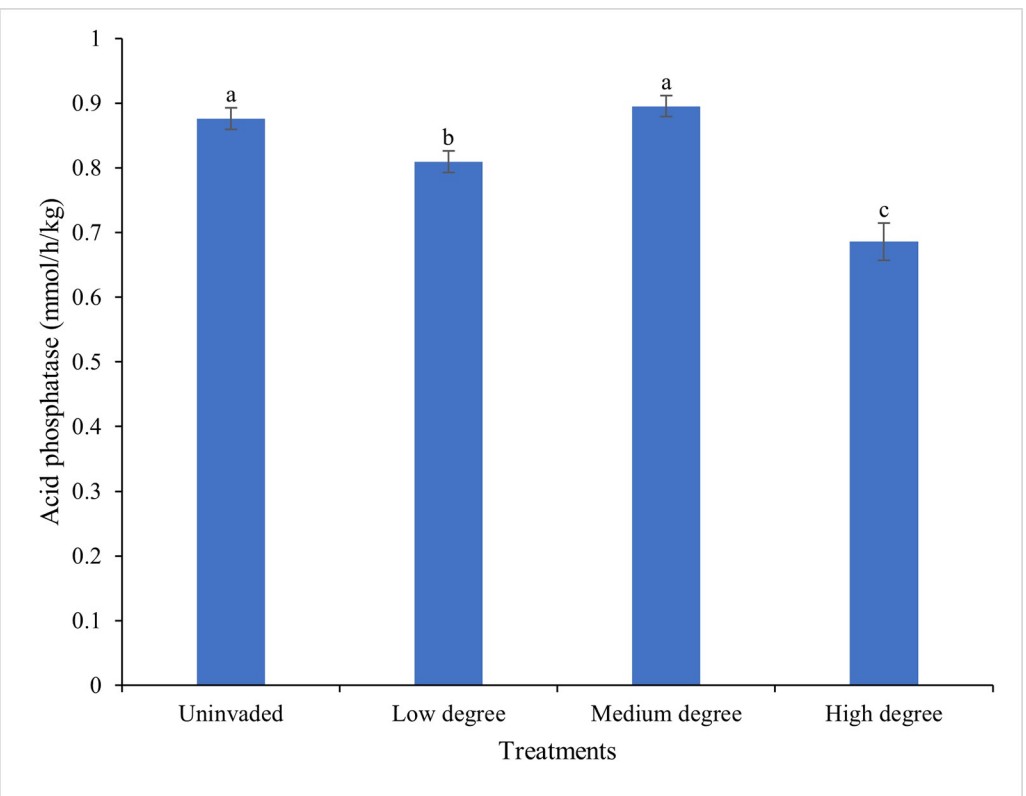

**Fig 8. Effect of different degrees of invasion on the acid phosphatase activity (mmol/h/kg) in the rhizosphere of *C. odorata*.** Data (Mean ± SD) with different letters are significantly different.

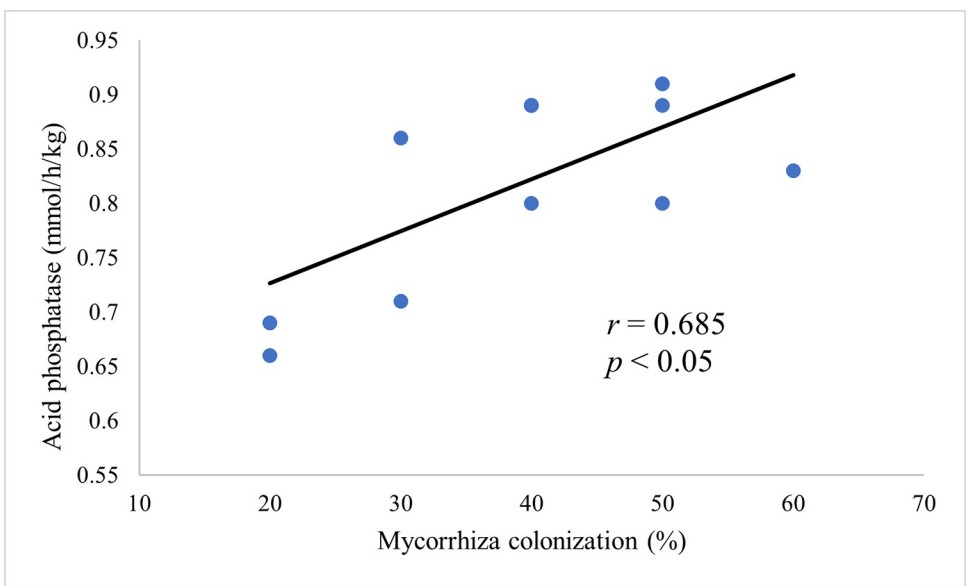

**Fig 9. Correlation between acid phosphatase activities in the rhizosphere of *C. odorata* and arbuscular mycorrhizal root colonization at different degrees of invasion.**

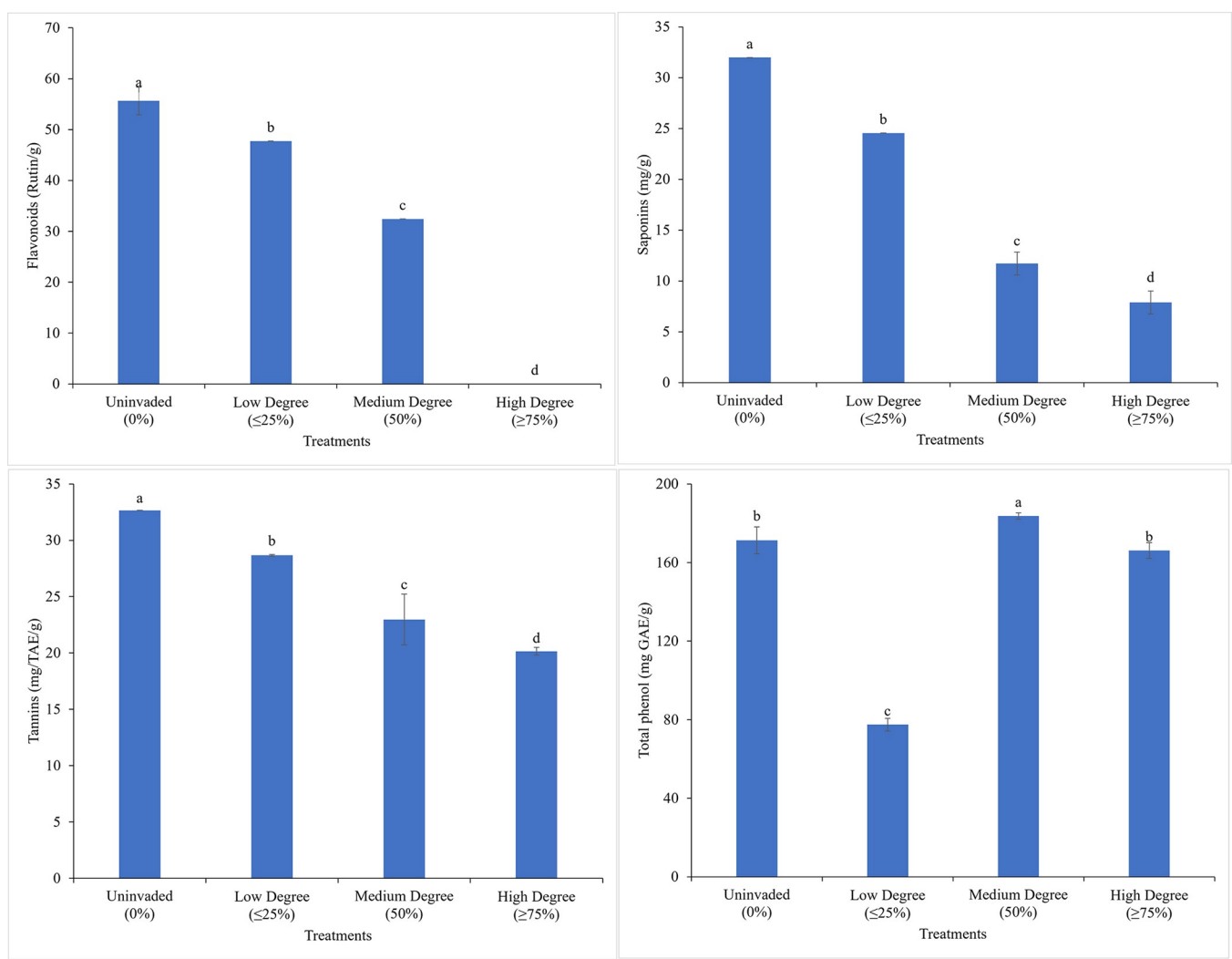

**Fig 10. Concentrations of secondary metabolites in rhizospheric soil samples at different degrees of *C. odorata* invasion.** Mean values with different letters indicate significant differences ($P<0.05$).

**Table 3. Correlation of saponins, flavonoids, tannins and total phenols.**

|  | Saponins | Flavonoids | Tannins | Total Phenols |
|---|---|---|---|---|
| Saponins | 1 | .900*** | .968*** | -0.336 |
| Flavonoids | .900*** | 1 | .909*** | -0.312 |
| Tannins | .968*** | .909*** | 1 | -0.298 |
| Total Phenols | -0.336 | -0.312 | -0.298 | 1 |

Values are significant at

*** = $P < 0.00$.

## 4. Discussion

### 4.1. Response of rhizosphere soil chemical properties to *C. odorata* invasion gradients

Invasive plants can modify soil chemical properties and microbial communities to establish themselves in new habitats [30,40]. The degree of *C. odorata* invasion had significant effects on soil chemical properties, which were essential indicators of soil quality for agricultural production and ecosystem maintenance. At the medium degree of invasion, soil P significantly increased, possibly due to high root colonization by AMF at the low degree of invasion. For instance, since AMF enhances phosphorus availability, especially in nutrient poor soils, Ngosong et al. [41] indicated that AMF plays a crucial role in increasing phosphatase activity and eventually P availability in nutrient poor soils. Therefore, the interactive effect of AMF and *C. odorata* play a crucial role in improving soil fertility. Earlier Studies have also shown that rhizosphere microorganisms such as bacteria and other fungi interacting with AMF did not only increased P availability but also increased P uptake, stimulating root growth, or excreting phytase, which broke down phytate [42]. However, excess soil P can lower P use efficiency and cause environmental risks such as eutrophication of aquatic systems thereby highlighting the need for careful management of plant invasive weed species such as *C. odorata* that can alter ecosystem and nutrient dynamics [43].

At the high degree of *C. odorata* invasion, there was a significant drop in soil P, K and Fe, accompanied by an increase in the microbial load of the rhizospheric soil samples. High microbial load can immobilize P, inhibit root growth and reduce plant P availability, leading to a drop in soil P content [3,21]. Phosphorus is essential for photosynthesis, respiration, energy storage and transfer, cell division and seed formation, and promotes early root formation and growth, improving crop quality [44]. Therefore, reduced soil P content caused by *C. odorata* invasion at low-degree invasion may increase its invasiveness in the area. This is because, the complex relationship between nutrient availability and plant invasiveness reduces nutrient levels creating a condition that favours the invasive species while disfavouring native species in that particular area [45,46]. A drop in the concentration of K in rhizospheric soils can have significant implications for plant growth and water use efficiency. Also, the effects of K in stress tolerance, for instance drought, and overall plant health cannot go unnoticed [47]. These effects may be minimal within a small range but become more pronounced after several decades of *C. odorata* invasion in the MCR. Potassium plays a crucial role in plant growth as an enzyme activator and helps regulate the plant's water use by controlling stomatal opening and closing [13,22]. It also maintains the balance of electrical charges at the site of ATP production, promotes the translocation of sugars for plant growth or storage in fruits or roots, is involved in protein synthesis, improves disease resistance, size of grains and seeds and quality of fruits and vegetables [1].

On the other hand, a drop in Fe concentration in rhizospheric soils can affect the activities of key enzymes such as catalase, peroxidase, cytochrome oxidase and the synthesis and maintenance of chlorophyll. Despite this being a pertinent issue, through the production of siderophores, and the release of organic acid anions by microorganisms, plant health especially in nutrient poor soils is maintained [44]. In contrast, a high degree of *C. odorata* invasion can significantly increase the concentration of soil Ca, which improves carbohydrate transfer and combines with anions, including organic acids, sulphates, and phosphates. This increase can also cause an elevation in soil C, N and organic matter content, which could be attributed to an increase in soil fungal load and a high bacteria count under a high invasion degree. Soil bacteria fix atmospheric nitrogen, increasing the accessible nitrogen pool available to plants in the

rhizosphere [12]. Furthermore, the effect of microorganisms in the elevation of soil organic matter generally improves soil structure and fertility as discussed by Coonan et al. [48].

Interestingly, *C. odorata* invasion did not significantly alter soil pH and moisture content at different degrees of invasion, contrary to some earlier findings on other invasive plants. Though these findings indicated a significant fluctuation in the soil pH and moisture content, the stability of soil moisture level and pH as observed in *C. ordorata* invaded areas suggests that this invasive species relies more on controlling soil nutrient dynamics than some chemical and physical soil properties such as pH and soil moisture respectively [49]. Alterations in soil pH and moisture facilitate the invasiveness of some species but are detrimental to the activities of microorganisms [49,50]. The mechanisms by which *C. odorata* invade may thus be different from alteration of soil pH and moisture, requiring further investigations. Presumably the effects of *C. odorata* invasion on soil chemical properties can have significant implications for plant growth, water use efficiency, and soil quality. It is crucial to continue investigating the mechanisms by which invasive plants modify soils and the implications for ecosystem maintenance, agricultural production and human well-being.

## 4.2. Dynamics of phosphatase enzyme activity

At a medium degree of invasion, there was a significant increase in acid phosphatase activities, complemented by an increase in soil P content and AMF colonization, which could have been attributed to improved P cycling [51]. This improvement of P cycling emanates from the release of organic acid from plant roots that solubilise other plant nutrients including phosphorus. This often occur to a greater extent in phosphorus deficient soils. Turner and Haygarth [52], reported that other soil properties such as pH, total N, organic P, clay content, microbial count and the amount of organic material present played crucial roles in phosphorus cycling. Furthermore, the interplay of these two complexes, soil chemical properties and phosphatase activity, is critical as some soil chemical properties such as organic matter increase enzyme activities through the enhancement soil microbial activity [53]. Further studies have emphasised the contribution of phosphatase-producing *Pseudomonas sp.* was also noteworthy in stimulating enzyme synthesis by intact roots and releasing inorganic phosphate from organic moiety and complex inorganic materials, thereby playing an essential role in the phosphorus cycle [16]. These studies have clearly indicated that *Pseudomonas sp.* can increase phosphorus availability through solubilization and mineralization processes that is fuelled by phosphatase activities in the plant rhizosphere. These bacterial species also played essential roles in phosphorus cycling by solubilizing organic and inorganic phosphates into available forms that support plant growth [17].

According to Wang et al. [13], phosphatase enzymes contributed to the cleavage of organic phosphorus, hence supplying organic phosphorus to soil solution by removing a phosphate group from a protein. Phosphatase ability to liberate phosphorus from organic compounds showcases their importance in nutrient cycling, especially in phosphorus poor soils. Interestingly, acid phosphatase activity was high in all degrees of invasion, suggesting its potential as a fallow crop and for the modification of soil biochemical processes. Furthermore, C. odorata playing the dual role as an invasive species and a potential soil amender highlights the need to carefully manage invasive species in all agricultural systems. Therefore, these finding indicated that *C. odorata* invasion can have some benefits in terms of soil fertility management, which could be explored further. Overall, understanding the mechanisms and implications of *C. odorata* invasion on soil processes and plant growth is crucial for addressing the challenges of invasive species management and promoting sustainable agricultural practices.

## 4.3. Dynamics of bacterial and fungal load in the rhizosphere

At a high degree of invasion, it was observed that *C. odorata* invasion did not significantly alter bacterial load but increased fungal load. The significant predominance of fungal species could be attributed to its numerous roles in altering soil chemistry as they have the ability to decompose organic matter and recycle plant nutrient in the soil. Furthermore, though the bacterial load was quite stable in this study, the statistical fluctuation in the fungal load could signify dynamics in the nutrient cycling processes [54]. This finding was intriguing, as the abundance and diversity of microorganisms in the soil played essential roles in maintaining soil fertility [54]. Moreover, microorganisms were recognised as potential indicators of some ecological stresses such as heat, moisture, salts, heavy metals, pesticides and pollution [55]. The bacterial count was high in all samples. The presence of gram-positive bacteria that tested positive for catalase activities indicated that a majority of the bacteria were strict aerobes or facultative anaerobes such as *Staphylococcus aureus*. Sizar et al. [56], revealed that *Staphylococcus* in the rhizosphere indicated a disruption of the natural microbiome community, leading to an increase in pathogens that caused food poisoning, toxic shock syndrome, septic arthritis, skin infections, scalded skin syndrome, pneumonia, abscesses, osteomyelitis, inflammatory disease, exfoliative toxin and endocarditis. High amounts of this bacterium in rhizospheric soils of *C. odorata* posed serious human health concerns.

It is, therefore, reasonable to presume that at low degrees of invasion, *C. odorata* might induce environmental stresses that could cause a significant drop in soil fungal load as suggested by Zhang et al. [57] who indicated most invasive plant species can secrete allelopathic compounds that hinders the growth of certain microorganisms such as fungi and bacteria, leading to reduce microbial abundance and diversity. This observation by Zhang et al. [57] is in line with the findings of this studies that showed a decline in fungal populations at lower invasion levels. This observation was also consistent with earlier findings that had reported altered microbial communities and ecological processes under invasive plant species [1,58]. The disrupted microbial communities and ecological processes could negatively impact soil fertility, plant growth and ecosystem functioning. Understanding the mechanisms and implications of *C. odorata* invasion on soil microbial communities could aid in developing effective strategies to manage invasive species and promote sustainable agricultural practices. It could also inform the conservation of biodiversity and ecosystem services.

## 4.4. *C. odorata* invasion enhanced Mycorrhizal colonization

It was observed that at a low degree of invasion, mycorrhizal colonization was significantly high, and this correlated positively with enzyme phosphatase activities suggesting that C. odorata facilitated AMF association with this invasive species thereby enhancing nutrient availability [41]. This finding was noteworthy, given that several studies indicated that soil microorganisms and mycorrhizal associations played crucial roles in improving soil fertility through their activities that involves extension of the root system's reach by the mycorrhiza arbuscules that proliferates soil aggregates increasing the feeder zone of the plant and allowing greater nutrient absorption especially in nutrient poor soils [12,59]. The symbiotic relationship between plants and mycorrhizal fungi was reported to enhance nutrient uptake and stimulate plant growth, particularly in nutrient-poor soils [8,9]. This suggested that the observed increase in mycorrhizal colonization and phosphatase activities could be attributed to the establishment of mycorrhizal fungi associations in response to the low degree of invasion. This was backed by a positive correlation between mycorrhizal root colonization and phosphatase activity that indicated an active role of AMF in mobilizing the phosphorus that emanates as a result of high phosphatase activity in the plant rhizosphere. The mechanisms and implications

of mycorrhizal associations and other soil microbial communities on soil fertility management and plant growth was crucial for addressing the challenges of sustainable agriculture and invasive species management [35,40]. This was clearly discussed by Bagyaraj et al. [59] who suggested that promoting plant and AMF associations can help maintain biodiversity and improve crop productivity.

## 4.5. *C. odorata* invasion gradients on the impact of the concentrations of secondary metabolites

Observations from the study indicated that low-degree invasion of *C. odorata* significantly increased the concentrations of saponins, flavonoids and tannins in the soil samples. This may have been attributed to the different invasion density of *C. odorata* that could have affected the concentration of these secondary metabolites. Furthermore, the complex relationship between different invasion densities and secondary metabolite concentrations suggests that alien plant species such as *C. odorata* may be manipulating its ecological vicinity to enhance its competitiveness in a new environment. Kato et al. [60] reported that invasive plant species can produce higher levels of secondary metabolites compared to native species to increase its probability to survive. Though root exudates were shown to impact soil chemistry, microbial populations and soil processes and defended the rhizosphere and root against pathogenic microorganisms [54], the invasion density could have been the key component responsible for the variable concentration of saponins, flavonoids and tannins in the soil. The significant increase in the soil concentration of saponins at low degree invasion suggested that *C. odorata* had important potential for leaching remediation, microbial remediation and phytoremediation, particularly with regards to heavy metal pollutants such as Cd, Zn, Cu and Pb. The mechanism that leads to phytohormone remediation by saponins involves the ability of saponins to bind to heavy metals and soil organic pollutants [61]. High soil concentrations of saponins can remove hydrophobic organic compounds and heavy metals from contaminated soils and water [62,63].

Additionally, the significantly high concentration of flavonoids recorded at low degree invasion suggested that *C. odorata* root exudates had antioxidant, anti-inflammatory, anti-cancer, anti-obesity, cardio-protective and neuroprotective activities. Flavonoids are known to protect plants from biotic and abiotic stresses, act as UV filters, provide drought resistance and aid in plant acclimatization [64,65]. Furthermore, the crucial role played by flavonoids in plant defence mechanisms influences not only the health of plants but also the health of soils by promoting microbial communities that enhance nutrient cycling [66]. The increased level of tannins recorded at low degree invasion was most probably linked to allelopathic responses and poor soil quality. At high degree of invasion, there was a significant drop in the concentration of total phenolic compounds from the rhizospheric soils, which most probably increased soil nitrogen and organic matter by reducing the complexation of proteins. This finding is consistent with earlier studies that reported elevated levels of soil nitrogen and organic matter in soils supplied with phenolic compounds, thus providing an insight into the repeated use of *C. odorata* as a fallow crop in several agroecological ecosystems [67,68].

The positive strong correlation between the secondary metabolites (saponins, flavonoids and tannins) indicated that the different degrees of invasion had a direct effect on the concentration of these metabolites as the increase in the concentration of one of the metabolites under the various invasion degrees led to an increase in another [69]. The strong positive correlations between these metabolites might have indicated a coordinated defence mechanism, where the production of one compound triggered the production of the others [70–72]. Also, the strong positive correlation could have been attributed to the fact that plants responded to

the attack by providing certain defence mechanisms that led to the increase in the concentration of one compound as a result of another [73]. Also, it was reported that tannins worked in synergy with saponins to increase their overall protective effects [69,74]. Therefore, the different plant metabolites work hand in hand suggesting a holistic interactive relationship where the presence of one metabolite enhances the synthesis and performance of the other, thereby strengthening the plant defence against threats.

## 5. Conclusion

The gradient of *Chromolaena odorata* invasion had varying effects on soil phosphorus, carbon and nitrogen in rhizospheric soils. It also altered soil fungal load and mycorrhizal colonization together with acid phosphatase enzyme activity and secondary metabolites exudates. *C. odorata* invaded uninvaded areas in mechanisms different from alteration of soil pH and moisture, but an interplay of changes in soil nutrients, microorganisms and release of secondary metabolites. This had varying effects on soil characteristics, with implications for soil quality and ecosystem conservation. These findings demonstrated the significant effects of *C. odorata* invasion on soil chemical properties, microbial communities, phosphatase activities and secondary metabolites at different gradients of the invasion process. They highlighted the importance of understanding the mechanisms by which invasive species modified soil properties and the impacted on microbial communities and plant growth to inform sustainable land management practices. The results emphasised the interdependence of different components of soil microbiota and the need for further investigation to understand the complex interactions between invasive species and soil micro-organisms. Low-degree invasion resulted in increased secretion of flavonoids, saponins and tannins, which provided insights into invasive species' potential benefits and drawbacks on the ecosystem and underscored the importance of continuing field investigation. Further work is needed to improve our understanding of the interplay between different degrees of invasion and soil chemical properties, the changes in microbial communities and the quantification of secondary metabolites for possible medicinal and agricultural purposes. Identifying microbial communities, characterizing various secondary metabolites, and monitoring soil changes influenced by *C. odorata* are crucial for effective ecosystem management. Understanding these ecological dynamics allows for precise and targeted strategies to mitigate the negative impacts of *C. odorata*, enhancing soil health and promoting microbial biodiversity. By elucidating the interactive effect between *C. odorata* and soil microbiomes, we can develop precise sustainable management practices that will go a great length to preserve native ecosystems and improve agricultural productivity. This integrated holistic approach will not only address immediate ecological concerns but will also foster long-term resilience against future invasions by invasive plant species.

## Supporting information

**S1 Data.**
(XLSX)

## Acknowledgments

The authors are grateful to the management bodies of the Agroecology Laboratory, University of Buea, Regional Biocontrol and Applied Microbiology Laboratory of IRAD, Yaoundé and Soil Microbiology Laboratory of the Biotechnology Centre of the University of Yaounde I, Cameroon.

## Author Contributions

**Conceptualization:** Lawrence Monah Ndam, Beatrice Ambo Fonge.

**Data curation:** Victor Nzengong Juru, Blaise Nangsingnyuy Tatah.

**Formal analysis:** Victor Nzengong Juru.

**Investigation:** Lawrence Monah Ndam, Beatrice Ambo Fonge.

**Methodology:** Victor Nzengong Juru, Lawrence Monah Ndam, Blaise Nangsingnyuy Tatah.

**Supervision:** Lawrence Monah Ndam, Beatrice Ambo Fonge.

**Validation:** Lawrence Monah Ndam, Beatrice Ambo Fonge.

**Visualization:** Lawrence Monah Ndam.

**Writing – original draft:** Victor Nzengong Juru, Lawrence Monah Ndam, Blaise Nangsing-nyuy Tatah, Beatrice Ambo Fonge.

**Writing – review & editing:** Victor Nzengong Juru, Lawrence Monah Ndam, Beatrice Ambo Fonge.

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
