## [Decision Letter · Decision Letter 0]

24 Jul 2024

PONE-D-24-20731Rhizospheric Soil Physicochemical Properties and Microbial Response to a Gradient of Chromolaena odorata Invasion in the Mount Cameroon RegionPLOS ONE

Dear Dr. Ndam,

Thank you for submitting your manuscript to PLOS ONE. After careful consideration, we feel that it has merit but does not fully meet PLOS ONE’s publication criteria as it currently stands. Therefore, we invite you to submit a revised version of the manuscript that addresses the points raised during the review process.

We look forward to receiving your revised manuscript.

Kind regards,

Md. Mahmudul Hasan, PhD

Academic Editor

PLOS ONE

A clean copy of the edited manuscript (uploaded as the new *manuscript* file).

4. We note that your Data Availability Statement is currently as follows: [All relevant data are within the manuscript and its Supporting Information files]

5. We note that Figure(s) 2 and 10 in your submission contain copyrighted images. All PLOS content is published under the Creative Commons Attribution License (CC BY 4.0), which means that the manuscript, images, and Supporting Information files will be freely available online, and any third party is permitted to access, download, copy, distribute, and use these materials in any way, even commercially, with proper attribution. For more information, see our copyright guidelines: http://journals.plos.org/plosone/s/licenses-and-copyright.

a. You may seek permission from the original copyright holder of Figure(s) 2 and 10  to publish the content specifically under the CC BY 4.0 license. 

Reviewers' comments:

Reviewer's Responses to Questions

**Comments to the Author**

1. Is the manuscript technically sound, and do the data support the conclusions?

Reviewer #1: Partly

Reviewer #2: Yes

Reviewer #3: No

2. Has the statistical analysis been performed appropriately and rigorously? 

Reviewer #1: I Don't Know

Reviewer #2: Yes

Reviewer #3: Yes

3. Have the authors made all data underlying the findings in their manuscript fully available?

Reviewer #1: No

Reviewer #2: Yes

Reviewer #3: Yes

4. Is the manuscript presented in an intelligible fashion and written in standard English?

Reviewer #1: Yes

Reviewer #2: Yes

Reviewer #3: Yes

5. Review Comments to the Author

Reviewer #1: The author kept saying physicochemical properties of rhizospheric soil in the manuscript but only chemical properties were determined and of course biochemical and microbiological properties of the rhizosphere soil. For your manuscript to qualify as "physicochemical", physical properties of soil have to be included. Such properties as bulk density, infiltration, particle density, mean weight diameter, porosity, aggregate stability, particle size distributions e.t.c. have to be included. I suggest therefore, the word "Physicochemical" should not be used as suggested in the reviewed manuscript uploaded herewith.

Reviewer #2: Review comments: PONE-D-24-20731

This study determined the effects of gradient densities of plant invasion on the soil properties, enzyme activities and population differences of microbial communities in field. The study showed many data of soil properties, enzyme activities, secondary metabolites and the bacterial and fungal population densities. These data analyzed in this manuscript are interesting. However, there were several shortcomings of the manuscript written in this style. I suggest this manuscript a major revision before publication.

General comments:

Introduction: paragraph 4 and 5 (line 71-88) could be rewritten before paragraph 2 (……Invasive species release root exudates……).

Methods

Line 136 to 138, soil samples collected with 5 cm of C. odorata rhizosphere from the four study areas. I think these samples are bulk soils but not rhizosphere soils (mm). Please check and improve.

Line 256, The phenolic components and flavonoids from soils collected were determined. Therefore, these could not be referred as root exudates in this study. Different invasion density of this plant could affect the composition contents of soil secondary metabolites. But whether these secondary metabolites were secreted by C. odorata were not determined.

Results

The photos were poorly generated by Excel. Please improve.

The results of contents of secondary metabolites affected by the gradient of plant invasion density were interesting. Correlation between them could be analyzed and improved in the discussion section.

Specific comments:

The calibration curves of secondary metabolites (Fig.3 to 6) could be listed in the supplied materials.

Some words were misspelled such as pH, nitrogen, organic matter, in Fig.7. Please check and improve.

Reviewer #3: Comments for Author,

Thank you for your submission. Even the paper is interessting, there are some issue to improve this paper.

Please obtained results are unclear. Please add your future recommandation end of conclusion remark.

Discussion is superficial. Please reconsider all obtained results and support with data analyses.

Best regards

6. PLOS authors have the option to publish the peer review history of their article (what does this mean?). If published, this will include your full peer review and any attached files.

Reviewer #1: **Yes: **Ibrahim A Aliyu (PhD)

Reviewer #2: No

Reviewer #3: No

---

## [Author Response · Author response to Decision Letter 0]

14 Sep 2024

Ndam Lawrence Monah 

Department of Agronomic and Applied molecular Sciences 

Faculty of Agriculture and Veterinary Medicine

University of Buea 

4th September, 2024

The editor-in-chief, 

PLOS ONE 

Dear Sir, 

 Responses to Reviewers` Concerns 

our manuscript No. [PONE-D-24-20731] - [EMID:b56eabaefbd680fb] entitled " Rhizospheric Soil Physicochemical Properties and Microbial Response to a Gradient of Chromolaena odorata (L) Invasion in the Mount Cameroon Region" refers. We thank you for the review of our article and the highlighted points to improve on its quality. The reviewers recommended a number of issues to be addressed for clarity and quality. Accordingly, we have addressed the points raised, revised the manuscript and made some rebuttal and clarifications where necessary. It is our hope that these revisions are satisfactory, and we look forward to your publication of the article. 

Meanwhile, the responses to reviewers’ comments are attached here below for your appraisal. 

Sincerely, 

Ndam Lawrence Monah 

Corresponding author 

 Responses to Reviewers’ comments 

Response:

Thank you for this comment. The manuscript has been edited and tailored to meet the PLOS ONE's style requirements.

Response:

Thank you for this observation. We have included a statement in the method section that reads as follows:

The study investigated an invasive plant species, regarded as a weed, in publicly accessible areas of Mount Cameroon through non-invasive observations and experiments. As it did not involve environmental manipulation or specimen collection for commercial purposes and aimed to enhance ecological understanding and management, no formal permits were necessary under local regulations.

The American Journal Experts (AJE) (https://www.aje.com/) is one such service that has extensive experience helping authors meet PLOS guidelines and can provide language editing, translation, manuscript formatting, and figure formatting to ensure your manuscript meets our submission guidelines. Please note that having the manuscript copyedited by AJE or any other editing services does not guarantee selection for peer review or acceptance for publication. Upon resubmission, please provide the following:

A clean copy of the edited manuscript (uploaded as the new *manuscript* file).

Response

The manuscript has been read again by all authors and the right language was used, and all spelling and grammatical mistakes corrected. Additionally, Prof. Njock Thomas Eku and Dr Ewane Divine of the University of Buea read and edited the manuscript for anomalies with regards to the right language, and all spelling and grammatical errors. A copy of the manuscript showing track changes of all the corrections has been provided. A copy of the clean manuscript has also been provided. 

4. We note that your Data Availability Statement is currently as follows: [All relevant data are within the manuscript and its Supporting Information files]

Response

The data used in this research is part of the data collected by a PhD student (Juru Victor, first author of the MS) pending the defense of his thesis. It is a tradition of the institution (University of Buea) that data be shared upon request, after a student has effectively defended his/her PhD thesis and submitted a copy of the thesis to the Library. Attached is a document from the institution on data sharing.

5. We note that Figure(s) 2 and 10 in your submission contain copyrighted images. All PLOS content is published under the Creative Commons Attribution License (CC BY 4.0), which means that the manuscript, images, and Supporting Information files will be freely available online, and any third party is permitted to access, download, copy, distribute, and use these materials in any way, even commercially, with proper attribution. For more information, see our copyright guidelines: http://journals.plos.org/plosone/s/licenses-and-copyright.

a. You may seek permission from the original copyright holder of Figure(s) 2 and 10 to publish the content specifically under the CC BY 4.0 license. 

Response

Thank you for this comment. Figures 2 and 10 have been removed from the manuscript.

ADDITIONAL COMMENTS FROM JOURNAL AFTER FIRST RESUBMISSION

IMPORTANT: PLEASE DO NOT REPLY TO THIS EMAIL

If you are unable to complete any points that are requested in this email, please explain why in the 'Enter Comments' tab of the online submission form prior to re-submitting your manuscript. This will let us quickly assess your response and send your manuscript to an Academic Editor as soon as possible.

PONE-D-24-20731R1

Rhizospheric Soil Chemical Properties and Microbial Response to a Gradient of Chromolaena odorata(L) Invasion in the Mount Cameroon Region

Dr Lawrence Monah Ndam

Dear Dr. Ndam,

We've checked your submission and before we can proceed, we need you to address the following issues:

1. In the online submission form, you indicated that "The data used in this research is part of the data collected by a PhD student (Juru Victor, first author of the MS) pending the defense of his thesis. It is a tradition of the institution (University of Buea) that data be shared upon request, after a student has effectively defended his/her PhD thesis and submitted a copy of the thesis to the library. Attached is a document from the institution on data sharing."

3. Uploaded as supplementary information.

Response

All relevant data are within the manuscript and its Supporting Information files.

2. We note that Figure 1 in your submission contain copyrighted images. All PLOS content is published under the Creative Commons Attribution License (CC BY 4.0), which means that the manuscript, images, and Supporting Information files will be freely available online, and any third party is permitted to access, download, copy, distribute, and use these materials in any way, even commercially, with proper attribution. For more information, see our copyright guidelines: http://journals.plos.org/plosone/s/licenses-and-copyright.

Response

Thank you for this comment. Figure 1 has been removed from the manuscript.

Reviewers' comments:

Reviewer's Responses to Questions

Comments to the Author

1. Is the manuscript technically sound, and do the data support the conclusions?

Reviewer #1: Partly

Reviewer #2: Yes

Reviewer #3: No

2. Has the statistical analysis been performed appropriately and rigorously?

Reviewer #1: I Don't Know

Reviewer #2: Yes

Reviewer #3: Yes

3. Have the authors made all data underlying the findings in their manuscript fully available?

Reviewer #1: No

Reviewer #2: Yes

Reviewer #3: Yes

4. Is the manuscript presented in an intelligible fashion and written in standard English?

Reviewer #1: Yes

Reviewer #2: Yes

Reviewer #3: Yes

5. Review Comments to the Author

Reviewer #1: The author kept saying physicochemical properties of rhizospheric soil in the manuscript but only chemical properties were determined and of course biochemical and microbiological properties of the rhizosphere soil. For your manuscript to qualify as "physicochemical", physical properties of soil have to be included. Such properties as bulk density, infiltration, particle density, mean weight diameter, porosity, aggregate stability, particle size distributions e.t.c. have to be included. I suggest therefore, the word "Physicochemical" should not be used as suggested in the reviewed manuscript uploaded herewith.

Response

Thank you for this keen observation. The word “Physicochemical” properties have been rephrased to “Chemical” Properties through out the manuscript as requested. 

Reviewer #2: Review comments: PONE-D-24-20731

This study determined the effects of gradient densities of plant invasion on the soil properties, enzyme activities and population differences of microbial communities in field. The study showed many data of soil properties, enzyme activities, secondary metabolites and the bacterial and fungal population densities. These data analyzed in this manuscript are interesting. However, there were several shortcomings of the manuscript written in this style. I suggest this manuscript a major revision before publication.

General comments:

Introduction: paragraph 4 and 5 (line 71-88) could be rewritten before paragraph 2 (……Invasive species release root exudates……).

Response

Thank you for this key comment to make the introduction better. In the introduction, paragraph 4 and 5 (line 71-88) have been rewritten before paragraph 2 as requested. 

Methods

Line 136 to 138, soil samples collected with 5 cm of C. odorata rhizosphere from the four study areas. I think these samples are bulk soils but not rhizosphere soils (mm). Please check and improve.

Response

Thank you for your observation. This has been revised as suggested for easy understanding by the readers and the word “rhizosphere” has been changed to “around the root zone of C. odorata” as that is the exact position where the soil was collected.

Line 256, The phenolic components and flavonoids from soils collected were determined. Therefore, these could not be referred as root exudates in this study. Different invasion density of this plant could affect the composition contents of soil secondary metabolites. But whether t

---

## [Decision Letter · Decision Letter 1]

3 Oct 2024

Rhizospheric Soil Chemical Properties and Microbial Response to a Gradient of Chromolaena odorata(L) Invasion in the Mount Cameroon Region

PONE-D-24-20731R1

Dear Dr. Lawrence Monah Ndam,

We’re pleased to inform you that your manuscript has been judged scientifically suitable for publication and will be formally accepted for publication once it meets all outstanding technical requirements.

Kind regards,

Md. Mahmudul Hasan

Academic Editor

PLOS ONE

Additional Editor Comments (optional):

Reviewers' comments:

Reviewer's Responses to Questions

**Comments to the Author**

1. If the authors have adequately addressed your comments raised in a previous round of review and you feel that this manuscript is now acceptable for publication, you may indicate that here to bypass the “Comments to the Author” section, enter your conflict of interest statement in the “Confidential to Editor” section, and submit your "Accept" recommendation.

Reviewer #2: All comments have been addressed

Reviewer #3: All comments have been addressed

2. Is the manuscript technically sound, and do the data support the conclusions?

Reviewer #2: Yes

Reviewer #3: Yes

3. Has the statistical analysis been performed appropriately and rigorously? 

Reviewer #2: Yes

Reviewer #3: Yes

4. Have the authors made all data underlying the findings in their manuscript fully available?

Reviewer #2: Yes

Reviewer #3: Yes

5. Is the manuscript presented in an intelligible fashion and written in standard English?

Reviewer #2: Yes

Reviewer #3: No

6. Review Comments to the Author

Reviewer #2: This study determined the effects of gradient densities of plant invasion on the soil

properties, enzyme activities and population differences of microbial communities in

field. This study was revised according to previous comments/suggestion. Although the photo generated by Excel was not perfect, this article could be accepted.

Reviewer #3: Author improved paper carefully, now the paper could be accepted for the publication. In addition to this, the authors addresed all.

7. PLOS authors have the option to publish the peer review history of their article (what does this mean?). If published, this will include your full peer review and any attached files.

Reviewer #2: No

Reviewer #3: No

---

## [Editor Report · Acceptance letter]

16 Oct 2024

PONE-D-24-20731R1 

PLOS ONE

Dear Dr. Ndam, 

I'm pleased to inform you that your manuscript has been deemed suitable for publication in PLOS ONE. Congratulations! Your manuscript is now being handed over to our production team.

Kind regards, 

on behalf of

Dr. Md. Mahmudul Hasan 

Academic Editor

PLOS ONE